# Re-Engineering Therapeutic Anti-Aβ Monoclonal Antibody to Target Amyloid Light Chain

**DOI:** 10.3390/ijms25031593

**Published:** 2024-01-27

**Authors:** Jingyi Bai, Xi Li, Jun Zhao, Huifang Zong, Yuan Yuan, Lei Wang, Xiaoshuai Zhang, Yong Ke, Lei Han, Jianrong Xu, Buyong Ma, Baohong Zhang, Jianwei Zhu

**Affiliations:** 1Engineering Research Center of Cell & Therapeutic Antibody, Ministry of Education, School of Pharmacy, Shanghai Jiao Tong University, Shanghai 200240, China; nicolebai2016@sjtu.edu.cn (J.B.); scarlet_lee@sjtu.edu.cn (X.L.); zhaoxiliunian@sjtu.edu.cn (H.Z.); yuanyuen@sjtu.edu.cn (Y.Y.); lwangph@sjtu.edu.cn (L.W.); xiaoshuaizhang@sjtu.edu.cn (X.Z.); keyong@sjtu.edu.cn (Y.K.); jianweiz@sjtu.edu.cn (J.Z.); 2Cancer and Inflammation Program, National Cancer Institute, Frederick, MD 21702, USA; jun.zhao@nih.gov; 3Jecho Biopharmaceutical Institute, Shanghai 200240, China; lei.han@jechoinst.com; 4School of Medicine, Shanghai Jiao Tong University, Shanghai 200025, China; janker.xu@gmail.com; 5Jecho Laboratories, Inc., Frederick, MD 21704, USA

**Keywords:** β-amyloid, amyloid light chain, amyloidosis, mAb, antibody design

## Abstract

Amyloidosis involves the deposition of misfolded proteins. Even though it is caused by different pathogenic mechanisms, in aggregate, it shares similar features. Here, we tested and confirmed a hypothesis that an amyloid antibody can be engineered by a few mutations to target a different species. Amyloid light chain (AL) and β-amyloid peptide (Aβ) are two therapeutic targets that are implicated in amyloid light chain amyloidosis and Alzheimer’s disease, respectively. Though crenezumab, an anti-Aβ antibody, is currently unsuccessful, we chose it as a model to computationally design and prepare crenezumab variants, aiming to discover a novel antibody with high affinity to AL fibrils and to establish a technology platform for repurposing amyloid monoclonal antibodies. We successfully re-engineered crenezumab to bind both Aβ42 oligomers and AL fibrils with high binding affinities. It is capable of reversing Aβ42-oligomers-induced cytotoxicity, decreasing the formation of AL fibrils, and alleviating AL-fibrils-induced cytotoxicity in vitro. Our research demonstrated that an amyloid antibody could be engineered by a few mutations to bind new amyloid sequences, providing an efficient way to reposition a therapeutic antibody to target different amyloid diseases.

## 1. Introduction

Protein aggregation diseases (commonly known as amyloidoses) are caused by pathologic processes with protein fibril deposition in such organs as the brain, heart, liver, pancreas, kidneys, and lungs, resulting in structural damage and dysfunction [1]. Amyloidosis can be classified based on the types of precursor proteins, as well as their localized or systemic deposition [1,2]. Systemic amyloidosis can be associated with protein deposits in any part of the body, such as systemic light chain amyloidosis due to the accumulation of immunoglobulin light chain amyloid fibrils. Most diseases, including Alzheimer’s disease (AD), Parkinson’s disease (PD), and prion (“mad cow”), affect the nervous system, leading to the loss of neuronal structure and function [1,2,3].

Alzheimer’s disease is one of the most devastating neurodegenerative diseases without effective therapies [4]. Several strategies have been developed to reduce Aβ production, inhibit aggregation, or directly enhance clearance. Using antibodies and their fragments has been a major approach [5,6,7]. Crenezumab is a fully humanized immunoglobulin isotype G4 (IgG4) anti-Aβ monoclonal antibody, which is currently in phase III study [8]. It is designed to bind multiple forms of Aβ (monomers, oligomers, fibrils, and plaques), especially those with high affinity to oligomeric forms. Crenezumab binds both Aβ1-40 and Aβ1-42, and its ability to block Aβ aggregation, promote Aβ disaggregation of oligomers, and protect neurons from oligomer-induced cytotoxicity has been demonstrated in vitro [9]. Unfortunately, crenezumab failed in clinical phase III in June 2022. We hope that crenezumab’s clinical trial could be continued with improvement, similarly to Aducanumab. Amyloid light chain (AL) amyloidosis, as an independent disease uncorrelated with AD, is the most common primary systemic amyloidosis [10,11,12,13] and is difficult to cure [14]. It is caused by insoluble protein fiber deposition of light chain amyloid precursor in different tissues induced by the clonal proliferation of plasma cells [15]. In contrast to the highly focused therapeutic research on Alzheimer’s disease, there are very few antibodies that could bind or inhibit AL amyloid aggregation to reverse or halt the course of the disease [14,16,17,18,19]. The leading monoclonal antibody, CAEL-101 (11-1F4), has just finished its phase 1a/b study in AL patients [20]. The structural details of the AL amyloids have been revealed by biological experiments [18,21], computational simulation [22], solid-state NMR [23], and recent cryo-EM experiments [24,25]. Loop flip at Pro8 of LC was previously proposed to lead to a β-sheet conformation by inserting residues 1–7 into the β-strands between residues 9–15 and 16–26 [18]. Partial unfolded dimers were suggested to be on the pathways of AL amyloid formations [21]. Based on the loop flip of the N-terminal region, a modeled structure of AL fibril with misfolded dimer was studied [22]. A solid-state NMR experiment indicated that AL amyloid may have a non-native structure [23]. However, five recent cryo-EM experiments of the λ light chain AL fibrils from cardiac amyloidosis patients have shown two totally different AL fibril folds [24,25,26,27]. The entire N-terminal is exposed on four fibril structures [24,26,27], while in Swuec et al.’s structure [25], only residues 8–16 are exposed.

Although amyloid aggregates differ considerably in their primary sequences, all share common features, such as a high degree of β-sheet content and the capability to bind the heteroaromatic dyes Congo red and thioflavin T (ThT) [28]. Theoretically, if all amyloids share a similar β-pleated sheet or other structural features, one may re-engineer an antibody that recognizes one amyloid species to recognize another one [29]. There are indeed reports that some antibodies do recognize different amyloids, for example CAEL-101 (11-1F4) can bind both amyloid light chain and Aβ, and the anti-prion antibody 15B3 detects toxic amyloid-oligomers [30]. This suggests that therapeutic amyloid-targeting antibodies can be repositioned to treat different aggregation diseases [31]. Most amyloid-targeting antibodies are specific [32]; however, a few mutations may alter their specificity range, narrowing or, as a first proof-of-principle step, broadening it.

In this study, we tested the above hypothesis. Our approach is to re-engineer known therapeutic amyloid antibodies to target different amyloid antigens by the computational screening of potential mutations. We selected crenezumab as a starting antibody, with amyloid light chain in systemic light chain amyloidosis as a target. Based on molecular dynamics simulations of antibody–AL fibril interactions, we computationally designed 30 mutation groups, each with six residues mutated from the wild-type (crenezumab), to target the amyloid light chain. Following the successful approach of CAEL-101 (11-1F4), which used an LN (1–30) peptide as its initial antigen and testing assay, we also used the LN peptide as the AL fibril model. Finally, 29 antibody mutants were constructed, and 3 of them were found to have nanomolar affinities to both Aβ oligomers and AL fibrils. The biophysical and biological activities of these were studied further. The three reengineered antibodies had been characterized by differential scanning fluorimetry (DSF), molecular weight, and peptide mapping. Among them, there was a novel antibody with high binding affinity and good bioactivity in vitro compared with crenezumab. Overall, our work demonstrated that an amyloid antibody can be engineered by a few mutations to bind new amyloid sequences, providing an efficient way to reposition a therapeutic antibody to target different amyloid diseases.

## 2. Results

### 2.1. Molecular Modeling and Simulation of Antibody-AL Binding

To prepare the antibody–AL fibril docking (Figure 1, Step 1), the structure of the crenezumab antibody was modeled by template-based homology modeling using the SWISS-MODEL server with the CreneFab structures as templates (PDB ID: 5kmv/5kna) [33]. Missing residues were modeled by template-based homology modeling using the server [34]. The structure of AL fibril was selected from our previous work with a partial misfolded protomer [22]. This structure was used to successfully identify the binding of a designed peptide to κ light chain amyloid fibril [35]. The potential binding interfaces between the AL fibril and the antibody were initially searched using the HADDOCK 2.1 web server [36], obtaining nine different complex structures. We ranked the binding scores using Rosetta Docking, and the key residue contacts and scoring energies are reported in Table 1. The docking poses obtained have large structural variations, and we chose the complex with the best binding score (Figure 2A and Table 1, top ranked with energy of −2.75 kcal/mol) for further refinement using molecular dynamics simulations (Figure 1, step 2).

To identify the essential interactions between AL fibril and Fabs, we calculated the contact frequency of Fab residues with AL fibrils (Figure 2 and Table 2) using conformations from the simulations. All atoms within 3 Å between AL fibril and the Fabs during the last 100 ns simulation were considered as input into PROTMAP2D [37], which can calculate the accumulated contact map by summing up all the frames during the simulations. We selected residues with contact frequency larger than 2 and examine their alanine scanning binding energy change (Table 2). We further chose six CDR residues with binding energy loss < 20 kcal/mol for further mutation screening to enhance the computational binding energy gain. We also used amino acid preferences obtained from our database analysis [38] to select possible mutations. Among the total 256 mutations examined by the GBMV (generalized Born with molecular volume) energy calculations, most mutations could achieve affinity gain of > 20 kcal/mol. We selected the 29 mutations with gain > 75 kcal/mol for further experimental tests (Figure 1, step 3).

### 2.2. Engineered Antibodies Bind Both Aβ Oligomers and AL

The antibody DNA sequence, including VH, CH, and signal peptide, around 1.3 kb, was synthesized and then cloned into pcDNA3.1(−) vector for expression. Similarly, the DNA sequence, including VL, CL, and signal peptide, around 700 bp, was cloned into a pcDNA3.1(−) vector and sequenced for verification. All clones were confirmed by DNA sequencing, and the constructed expression vectors were separately named wild-type, M1, M2, M3, … M30. Finally, an expression vector of 29 mutation types and wild type were constructed successfully, except M17.

The antibodies were expressed in CHO-S cells and purified by Protein A chromatography. The purified products were analyzed by 10% SDS-PAGE, and the molecular weight was about 170 KD in non-reducing SDS-PAGE (Figure 3A), and the heavy chain and light chain were 55 KD and 25 KD, respectively (Figure 3B), in reducing SDS-PAGE.

ELISA analyses were performed to examine the affinities of all engineered antibodies to the Aβ42 oligomers and AL fibrils. All mutants have decreased binding affinity to Aβ42 oligomers in comparison with wild-type crenezumab (Figure 4A upper). Five engineered mutants (M2, M5, M10, M29, and M30) maintained a relatively higher level of binding to Aβ42 oligomers than the other mutants. Similar ELISA screening methods had been used for the binding of all antibodies to AL fibrils (Section 4.4). There was binding affinity between wild-type crenezumab and AL fibrils. While not all computationally engineered mutants have improved affinities to AL fibrils, 12 mutants (M5, M7, M8, M9, M10, M11, M15, M16, M18, M19, M29, M30) bind AL better than the wild-type crenezumab (Figure 4A lower).

Based on the ELISA assay results, eight engineered antibodies were screening out for further binding affinity verification using a surface plasmon resonance (SPR)-based assay on Biacore T200 instruments. The wild-type crenezumab included in this assay was used as control. The affinity of M5 and the wild-type crenezumab to Aβ42 oligomers and AL fibrils measured using surface plasmon resonance (SPR) is shown in Figure 4B. The further affinity of M5, M10, M29, and the wild-type crenezumab to Aβ42 oligomers and AL fibrils was also measured using surface plasmon resonance (SPR), and the result is shown in Figure 4C. The others’ original sensorgrams to Aβ42 oligomers and AL fibrils measured using SPR are shown in Appendix A, respectively. It is shown in Table 3 that the measured KD values generally reflected the trend observed in ELISA assays. For binding to Aβ42 oligomers, the KD value of wild-type crenezumab was 1.53 nM, which was close to the reported range of 3.0–5.0 nM to Aβ monomers and 0.4–0.6 nM to Aβ oligomers [39]. The single mutation in the crenezumab CDR region can cause a 2–10 times decrease of binding affinity [39]; therefore, it is not surprising that our engineered antibodies (with six mutations) are around 10 times weaker in binding Aβ oligomer.

The wild-type crenezumab had no binding to AL fibril in the SPR measurement. It was different from the ELISA assay. In comparison, the mutants engineered to target AL fibril had strong binding to AL fibrils in the low nanomolar range. According to the affinity to both Aβ42 oligomers and AL fibrils, M5, M10, and M29 were selected for further biophysical/biological studies.

### 2.3. Characterization of Engineered Aβ and AL Binding Antibodies

To confirm the exact amino acid sequence of purified M5, M10, and M29, we used LC-MS with support by Jecho Biopharmaceuticals Co., Ltd. (Tianjin, China) to examine the intact heavy chain and light chain molecular weights, and peptide mapping. Mass spectra of M5, M10, and M29 intact molecular weight are shown in Figure 5A, while those of M5, M10, and M29 heavy chain and light chain are shown in Appendix A. The intact molecular weight and heavy chain and light chain molecular weight of M5, M10, and M29 are reported in Table 4. These data essentially proved that the mAbs of M5, M10, and M29 were constructed and expressed successfully.

Peptide mappings were further analyzed by LC-MS for M5, M10, and M29. Three types of enzymes (trypsin, chymotrypsin, Glu-C) were used in peptide mapping analysis. As shown in Appendix A, the peptide digested by the three enzymes was completely separated in the chromatogram. In the subsequent LC-MS analysis, each digested peptide fragment was successfully identified by accurate mass, and some were further confirmed by their MS/MS fragment ions when necessary, including the N-terminal and C-terminal peptides of the heavy and light chains (Figure 5C). The sequence coverage of M5 and M29 was 100% for both heavy and light chains, 99% for the heavy chain of M10, and 100% for the light chain.

Thermal stability can directly reflect the conformational stability of an antibody. NanoDSF was performed using NanoTemper^®^ Technologies’ Prometheus^TM^ instruments (Prometheus NT.48) for comparison of the thermal stabilities between different antibodies. Thermal unfolding profiles (thermogram) of M5, M10, M29, and the wild type by NanoDSF are showed in Figure 5B. Data from Figure 5B and Table 5 show that the Tms of M5, M10, and M29 were 2–4 degree lower than that of the wild type, which indicated the lower thermal stability (conformational stability) of the engineered mutants. In Figure 5B and Table 5, data also show that M5, M10, and M29 had lower colloidal stability due to temperature since their Tagg onsets were 2–3 degrees lower than that of the wild type. While the Tagg scatting signal also demonstrated that M29 had a much lower aggregation intensity (a qualitative measure for the overall degree of aggregation) than the wild type (Table 5), M29 showed increased colloidal stability. Previous studies have also demonstrated that mutations in CDR regions affect the folding (conformational) stability [40,41,42] and colloidal stability (solubility) [43,44] of antibodies.

### 2.4. Bioactivity of Engineered Antibodies

Previous studies showed that Aβ oligomeric species were toxic species [45,46,47]. To compare the effects of M5, M10, and M29 with that of the wild type on Aβ42-oligomer-induced cell toxicity, we tested Aβ42 oligomers’ bioactivity by CCK-8 (Cell Counting Kit-8) assay on SH-SY5Y human neuroblastoma cell line. Figure 6A shows that cell viability significantly decreased when incubated with 20 μM Aβ42 oligomers (~40% cell viability) in an Aβ42-oligomers-concentration-dependent manner (100% cell viability without the oligomers), indicating toxicity of Aβ42 oligomers. We chose the concentration of 5 μM Aβ42 oligomers for cell toxicity evaluation, and incubated with M5, M10, M29, and wild-type crenezumab at different concentrations at 37 °C for 48 h in order to compare their reversing effects on the toxicity (Figure 6B). Samples co-incubated with the wild type, M5, M10, and M29 showed that the wild type had a better performance in preventing Aβ42-oligomer-induced cell toxicity than the engineered mutants. M5 had slightly better prevention than M10 and M29 at 0.25 and 0.5 μM concentrations. However, with increasing of the antibody concentration, all three mutants show similar, and dose-dependent, effects.

ThT (thioflavin T) dye exhibited a strong fluorescence intensity enhancement at an excitation wavelength of 450 nm and emission at 482 nm upon binding to amyloid fibrils [48]. Unbound ThT was essentially nonfluorescent at these wavelengths, which permitted its inclusion in the reaction solution without interfering with the measured fluorescence signal. We examined the inhibition effects of M5, M10, M29, and crenezumab on AL fibrils’ formation by monitoring ThT fluorescence. AL peptide solutions were subjected to constant rotational agitation at 37 °C for 72 h, and incubated with different concentrations of M5, M10, M29, and crenezumab (2, 1, 0.5, 0 μM). Amyloid formation was then assessed using ThT, and no increase of ThT fluorescence of antibodies incubated without AL fibrils could be observed compared to the control reactions. The effects of antibodies on AL fibrils formation are shown in Figure 6C. At low antibody concentration (0.5 μM), M29 and M10 surprisingly increased ThT fluorescence. In the presence of M5 and wild-type crenezumab at high concentration (2 μM), AL fibrillogenesis could be effectively inhibited. On the other hand, at lower antibody concentration, the wild type could not inhibit AL fibril formation.

AL fibrils has been confirmed to have strong cytotoxicity in cardiomyocytes [49]. In order to verify the reversal effect of M5 on the cytotoxicity induced by AL fibrils, we performed MTT assays in the AC16 cell line. Figure 6D shows that after 96 h incubation in vitro, the AL samples displayed a significant decrease in cell viability (~40% cell viability, 100% cell viability without AL fibrils). AL samples being co-cultured with different concentrations of M5 significantly alleviated the cytotoxicity, and cell viability rose to ~75% with the presence of 0.2 μM M5. The result indicated that M5 has a significant reversal effect on the cytotoxicity of AL fibrils in vitro in a dose-dependent manner.

## 3. Discussion

In the present study, helped by structure-based computational design, we successfully repurposed an Aβ-peptide antibody to bind AL protein deposits. Six mutations were introduced in CDR regions of crenezumab. We constructed the expression vectors for transient transfection into CHO-S cells, and the 29 mutants and wild-type crenezumab were prepared and analyzed for their binding affinity and biological activity to both β-amyloid oligomers and AL fibrils. ELISA and SPR assay results show that many mutations have increased their binding affinity to AL fibrils compared to the wild type. While the ELISA assay is not able to measure affinity accurately, and we only used it as a screen for any binding activity, the subsequent SPR assay may provide better quantitative insights. We noted that the binding of bidentate antibodies to immobilized antigen will likely result in cooperative binding of the two Fabs, and cleavage of the antibodies to monodentate Fabs should be carried out in the future to calculate affinity. Nevertheless, we found that M5, M10, and M29 also retained partial binding affinity to Aβ42 oligomers. In the Aβ42-oligomer-induced cell toxicity assay, when samples of 5 μM oligomers were added to SH-SY5Y cell culture, the cell viability decreased to 70%, and the addition of M5, M10, and M29 annulled the toxicity of the peptide on cell viability in a dose-dependent manner. At the concentration of 0.5 μM, the addition of M5 and M10 could inhibit Aβ42-induced cytotoxicity, indicating remarkable biological activity on the β-amyloid oligomers. Here, our aim is to test a combination of computational and experimental approaches to adjust the specificity and affinity of the antibody, specifically to repurpose an AD-targeting antibody toward other amyloidogenic diseases. Repurposing has proven an effective strategy in small-molecule drug discovery in part due to the demonstrated low toxicity and availability [31,50]. However, the common amyloid motif may still present a concentration and thus toxicity challenge, as we indeed observe here in the recognition of Aβ by the crenezumab mutants. This emphasizes the requirement for high specificity, which may still need further improvement.

Since the Abeta and LC have different sequences and overall fibril structures, their interactions are totally different. Comparing the antibody residues with high frequency in antigen contact of Abeta-antibody [51] and the current LC-antibody (Table 2), we see only four residues appear that recognize both Abeta and LC (Glu1H, Ser31H, Ser53H, and Ser32L). These results demonstrated the difference of antibody–amyloid recognition and antibody–globular protein recognition. The antibody–globular protein recognition often relies on specific interactions, in both linear and conformational epitopes. However, the antibody–amyloid recognition relies mostly on the overall amyloid conformation and topology shape.

Repurposing an antibody by minimum mutations has huge advantages compared to forming a new antibody from scratch. The antibodies in the late stages of development (clinical trials) have been mostly optimized for their stability and toxicity. While the mutations introduced would disturb the previously optimized antibodies, the re-adjustment (or re-optimization) of the slightly perturbated antibody is still much easier than optimization of new antibodies from scratch.

Antibodies can recognize conformational features of β-amyloid oligomers [52,53], and some antibodies were previously shown to be capable of binding not only to Aβ fibrils, but also to amyloid fibrils generated from other proteins. As we observed here, at higher concentration, wild-type crenezumab had an inhibition effect of AL fibrillogenesis, although it had relatively weak binding affinity to AL fibrils. This result indicated that crenezumab already has the potential to recognize the general amyloid conformational epitopes. Given the potential cross-reaction of antibodies to several forms of amyloids, it is very important for antibody engineering to be able to control the specificity and affinity towards a specific amyloid. Our results demonstrated that we may achieve several goals. The first is to completely switch specificity from one target to another, as in the case of M11, M15, and M18. The second is to add new affinity to a new target while maintaining the binding ability to the original target, as is the case of M5. While Alzheimer’s disease and systemic amyloid light chain are different diseases, it is of great interest to develop a bi-specific antibody to fight against protein aggregation diseases. For example, in future development, one may consider engineering an antibody to both Aβ and tau protein in Alzheimer’s disease. Other bi-specific targets are β-Amyloid and human islet amyloid polypeptide (hIAPP), which is implicated in type 2 diabetes (T2D). Increasing evidence suggests that AD and T2D may be correlated with each other [54].

The structure of the N-terminal region of the AL fibril could take difference conformations. The residues 1–14 may be highly flexible with all of the N-terminal exposed on the fibril surface [24], while only residues 8–16 are exposed for antibody binding in Swuec et al.’s structure [25]. Our current antibody bioactivities results confirmed our previous modeling of the AL fibril, which has the common U-turn-like β-sheet motif in β-amyloid fibrils [22]. It is interesting to note that our computational designs were based on the κ light chain variable domain (107 residues) dimer, which has a slightly different sequence from the experimental used LN (1–30) peptide in the Aβ42 oligomers bioactivity test. The LN peptide amyloid has served as a representation of a general light chain amyloid model for a long time. However, the LN peptide amyloid and overall AL amyloid may differ in both structural and sequential aspects. As can be seen in Figure 1, twelve of thirty residues of N-terminal residues of AL fibril and LN (1–30) peptide are different. Structurally, with all residues beyond the N-terminal region, the amyloid structures of the κ light chain variable domain and 30-residue-long LN peptide should be morphologically different. Still, our designed antibodies are able to bind LN peptide amyloid with nano-molar affinities, indicating that the N-terminal region of the light chain amyloid indeed shares structural similarity with the LN peptide amyloid, which has served as a representation of a general light chain amyloid model for a long time.

The abilities of our designed antibodies to bind the LN peptide amyloid with non-native immunoglobin fold is important. It is known that the β2-microglobulin fibril (another Ig-domain native state) shows a non-native arrangement of β-strands (68), and light-chain-derived AL amyloid has a non-native structure (69). While in the future we are going to test light-chain-derived AL amyloid directly, the current results of LN peptide amyloid binding indicate that our designed antibodies are not targeting native Ig fold; rather, they recognize the loop-flip feature of N-terminal residues.

## 4. Materials and Methods

### 4.1. Computational Antibody Re-Engineering 

The computational and experimental steps are listed in Figure 1. We have previously simulated crenezumab [55] and constructed a structural model of the AL fibril [23]. The sequence of the AL fibril is based on the κ light chain variable domain: D^1^IQMTQSP^8^SSLSASV^15^GDRVTITCQASQDISDYLIWYQQKLGKAPNLLIYDASTLETGVPSRFSGSGSGTEYTFTISSLQPEDIATYYCQQYDDLPYTFGQGTKVEIKR^107^. The structure loop-flip2 [22] was used as the target structure of the AL fibril to dock with crenezumab and in subsequent MD simulations. The epitope of AL fibrils is at Pro8-Val 15 [22], while the paratopes of crenezumab were set to the CDR loops. This information was given to the HADDOCK server [36]. The fibril was set rigid, and the conformations of the CDR loops of Fabs were taken from the crystal structures (i.e., 26–37, 55–58, 95–102, 245–253, 270–277, 318–320), were set flexible. Analysis of the final 200 crenezumab-AL fibrils HADDOCK models resulted in nine clusters. To further refine the Fabs-AL fibril complexes, the HADDOCK poses were locally perturbed by Rosetta docking [56]. Poses with larger cluster size, lower Z-score, and total_score/I_sc were considered promising candidates.

In the molecular dynamics simulations, the N- and C-termini of the VL protomers and the antibody were NH^3+^ and COO^−^ groups, respectively. The conserved intra-domain disulfide bonds were constructed. The systems were then solvated by TIP3P water molecules, and sodium and chlorides were added to neutralize the system and to achieve a total concentration of ~150 mM. The resulting systems were energy minimized for 5000 conjugate gradient steps, with the protein fixed and water molecules and counterions allowed to move, followed by an additional 5000 conjugate gradient steps, where all atoms could move. In the equilibration stage, each system was gradually relaxed by performing a series of dynamic cycles, in which the harmonic restraints on the proteins were gradually removed to optimize the protein–water interactions. In the production stage, all simulations were performed using the NPT ensemble at 310 K. All MD simulations were performed using the NAMD software version 2.12 [57] with CHARMM36 force field [58]. MD trajectories were saved every 2 ps for analysis.

To evaluate the binding energy between the wild-type Fab and the AL fibril, the trajectory for each bound and apo system was extracted from the last 20 ns of explicit solvent MD without water molecules and ions. To evaluate the binding energy between Fab mutants and AL fibril, the trajectories from wild-type Fab and AL fibril complex were used. The side chains of the residues that were mutated were removed and rebuilt. The rotamers of mutated residue side chains were optimized using PyMoL. The solvation energies of all systems were calculated using the generalized Born method with molecular volume (GBMV) [59] after 500 steps of energy minimization and 5 ns short MD simulation to relax the local geometries caused by the thermal fluctuations that occurred in the MD simulations. In the GBMV calculation, the dielectric constant of water is set to 80, and no distance cutoff is used. The binding energy between Fab and the AL fibril was calculated by $\langle E_{bind}\rangle =\langle E_{complex}\rangle -\langle E_{Fab}\rangle -\langle E_{A\beta }\rangle$.

### 4.2. Genes Cloning and Expression Vector Construction

The DNA sequences of wild type (crenezumab) and 30 mutation types were determined according to the above methods. VH, CH, VL, and CL DNA fragments of wild and mutation types, including the signal peptide, were amplified from the sequences synthesized by GenScript Corporation (Nanjing, Jiangsu, China). The full length of light chain and heavy chain DNA sequences of the wild and mutation types were respectively cloned using overlap PCR with the above DNA fragments as templates. The plasmid pcDNA3.1(−) was digested with *Nhe*I and *Hind*III (Takara Biomedical Technology, Beijing, China) before the DNA sequences were cloned into linearized pcDNA3.1(−). All clonings were confirmed by DNA sequencing, and the constructed expression vectors were separately named wild-type, M1, M2, M3, … M30. Finally, the expression vector of 29 mutation types and the wild type were constructed successfully, except M17.

### 4.3. Expression and Purification of Wild and Mutant Types

Each expression vector was prepared using the Endo-free Plasmid Maxi Kit (Omega Bio-Tek, Norcross, GA, USA) following the manufacturer’s instructions. The light chain and heavy chain plasmids were co-transfected into CHO-S cells. Prior to transfection, cells were centrifuged before being re-suspended at a cell density of 3 × 10^6^ cells/mL in 200 mL CD-CHO medium supplemented with 4% (*v*/*v*) Glutamax (Thermo Fisher, Life Technologies, Carlsbad, CA, USA). HC and LC plasmids were diluted with DMEM medium (Thermo Fisher, Life Technologies, Carlsbad, CA, USA) to a concentration of 40 μg/mL (LC:HC = 4:1, *w*:*w*) and mixed with 25 KD polyethyleneimine (PEI; Polysciences, Warrington, PA, USA) (DNA:PEI = 1:2, *w*:*w*) for incubation at room temperature for 15 min. Then, the mixture of DNA and PEI was added to the cell. Then, the transfected cells were cultured at 125 rpm at 37 °C. Supernatant was taken for purification or analysis when the cell viability decreased under 50%. Antibodies were purified using a Protein A column (GE Healthcare, Waukesha, WI, USA) individually, and the protein purities were determined by SDS-PAGE. The protein concentration of the samples was measured by BCA protein assay kit (Beyotime Institute of Biotechnology, Nantong, Jiangsu, China). 

### 4.4. Preparation of Aβ Oligomers and AL Amyloid Fibrils

Aβ42 synthetic lyophilized peptide (DAEFRHDSGYEVHHQKLVFFAEDVGSNKGAIIGLMVGGVVIA, 96.71%; GL Biochem, Shanghai, China) was dissolved in 1,1,1,3,3,3-hexafluoro-2-propanol (HF-IP, 99%; Macklin, Shanghai, China) at a concentration of 200 μM in 150 μL aliquots in order to obtain monodispersed peptide preparations [60] and air dried to remove HFIP, and the peptide film was stored at −80 °C until use. Aliquots of peptide film were dissolved with dry dimethyl-sulfoxide (DMSO, 99.7%; Macklin, Shanghai, China) at a concentration of 5 mM and further diluted in phosphate-buffered saline (PBS) to a final concentration of 100 μM peptide. Aβ42 oligomers were prepared by incubation at 4 °C for 24 h for affinity and in vitro bioactivity studies.

Lyophilized water-soluble light chain amyloid peptide (DIVMTQSPDSLAVSLGERATINCKSSQSVL, 95.57%; GL Biochem, Shanghai, China) was prepared in PBSA at a concentration of 1 mg/mL, before the solution was passed through a 0.2 μm pore-sized filter to remove formed aggregates. A volume of 2 mL solution was placed in a 15 mL polypropylene tube (Corning Incorporated, Corning, NY, USA) and shaken at 225 rpm at 37 °C for 2 weeks, as described in previous study [61], after which the components were harvested by centrifugation (20,800× *g* for 25 min), sonicated, and stored at −80 °C.

### 4.5. Affinity Screening by ELISA

Indirect ELISA was used to determine the antigen-binding activity of the 29 mutation types. Aβ42 oligomers or LC amyloid fibrils were coated overnight at a concentration of 500 ng/mL at 4 °C on the 96-well ELISA microplates (100 μL/well). Unspecific sites were blocked for 1 h at 37 °C with PBS buffer containing 0.05% Tween 20 (PBS-T) and 1% bovine serum albumin (BSA). After washing with PBS-T, 100 μL of 8.1 μg/mL antibodies of wild-type and different mutation types were added to the plates followed by incubation at 37 °C for 2 h. The plate was washed with PBS-T before 100 μL of horse radish peroxidase (HRP) conjugated donkey anti-human IgG (Jackson ImmunoResearch, West Grove, PA, USA) was added to each well, and the plates were incubated at 37 °C for another 1 h. The plate was then washed with PBS-T and incubated with 100 μL of 3,3′,5,5′-Tetramethylbenzidine (TMB) at room temperature in the dark for 20 min. The reaction was stopped by 2 M H_2_SO_4_, and optical density was determined at 450 nm and 630 nm (OD_450–630_) using an ELISA plate reader (Tecan, Männedorf, Switzerland).

### 4.6. Antibody Affinity Measurement by SPR

Biacore T200 instruments (GE Healthcare, Waukesha, WI, USA) were used to monitor binding interactions via SPR in order to compare the affinities of mAbs. AL fibrils and Aβ42 oligomers were immobilized using CM5 amine-coupling at a flow rate of 10 μL/min in 10 mM sodium acetate buffer (pH 3.8 and pH 4.0, respectively). The sensor surface was activated with a 7 min injection of 50 mM N-hydroxysuccinimide with 200 mM 1-ethyl-3-(3-dimethylamino-propyl)-carbodiimide hydrochloride. Then, 0.1 mg/mL of AL fibrils and Aβ42 oligomers was injected to reach the target level of 1000 RU and 500 RU, respectively. Then, the surface was blocked with 1 M ethanolamine (pH 8.5). Reference flow cells were prepared without protein. All binding measurements were performed at 25 °C in 10 mM phosphate buffer with 137 mM NaCl, 2.7 mM KCl, 0.05% (*v*/*v*) Tween-20, pH 7.5. Two-fold serial dilutions of antibodies were sequentially injected and analyzed at 30 μL/min. The association was set to 120 s, and the dissociation was set to 1200–3600 s for different antibodies. The regeneration was set to 30 s, by using 10 mM NaOH for AL fibrils and glycine-HCl (pH 1.5) for Aβ42 oligomers. Double reference subtractions were made to eliminate injection noise and data drift. Affinity and binding kinetic parameters were determined by global fitting to a Langmuir 1:1 binding model within the Biacore (GE Healthcare, Waukesha, WI, USA) Evaluation software (Biacore insight evaluation 3.0.12.15655).

### 4.7. Thermal Stability Analysis

The thermal stabilities of the antibodies were analyzed by Prometheus^TM^ NT.48 instruments (NanoTemper^®^ Technologies, Beijing, China) using nanoDSF, an advanced differential scanning fluorimetry technology. The antibodies were resuspended in PBS to obtain protein solution, followed by completely filling one capillary from this solution and placing it on the capillary tray. After starting a new session of the PR.ThermControl software version 2.1.1, go to Melting Scan and prepare a run with settings of 1.0 °C/min, 20–95 °C, 10% Excitation Power, and start the measurement.

### 4.8. Molecular Weight Analysis and Peptide Mapping 

A total of 200 μg of each mAb was concentrated using ultrafiltration centrifugal tubes and diluted with 50 mM ammonium bicarbonate to approximately 5 mg/mL. Then, 1 μL of PNGase F (NEB, Ipswich, MA, USA) was added to each sample in order to release the N-glycans, before the mixtures were incubated at 37 °C for 24 h to obtain the deglycosylated samples. After PNGase F treatment, 1 μL samples were mixed with 49 μL water and marked as unreduced samples for intact molecular weight analysis, and 1 μL samples were mixed with 49 μL 6 M guanidine hydrochloride (Sigma-Aldrich, St. Louis, MO, USA) and reduced with 5 μL 0.5 M DTT (Sigma-Aldrich, St. Louis, MO, USA) at 56 °C for 30 min for heavy chain and light chain molecular weight analysis. For peptide mapping, a total of 150 μg of each mAb was reacted with DTT and IAM (Sigma-Aldrich, St. Louis, MO, USA) in order to open the disulfide bond and alkylate the free sulfydryl. After the samples were divided into three copies respectively, each copy was digested with trypsin (Promega, Madison, WI, USA), chymotrypsin (Sigma-Aldrich, St. Louis, MO, USA), and Glu-C (Sigma-Aldrich, St. Louis, MO, USA) at 1:20 (enzyme:antibody, w:w) and incubated at 37 °C for 16 h. Then, 15% formic acid was added until pH was below 3.0 to stop the reaction. Waters ACQUITY UPLC H-Class system (Waters Corporation, Milford, MA, USA) and Waters ACQUITY UPLC Protein BEH C4 column (1 mm × 100 mm, 1.7 μm, Waters), C18 column (2.1 mm × 100 mm, 1.7 μm, Waters), were used for molecular weight analysis and peptide mapping.

### 4.9. Aβ42 Oligomers Bioactivity Measured by Toxicity on SH-SY5Y Cells

The SH-SY5Y human neuroblastoma cell line (American Type Culture Collection, ATCC) was maintained in DMEM medium (Thermo fisher, Life Technologies, Carlsbad, CA, USA) with 10% fetal bovine serum (Gibco, Carlsbad, CA, USA) and 1% antibiotics (Gibco, Carlsbad, CA, USA) in 5% CO_2_ at 37 °C. The cells were seeded at a density of 5000 cells/well in 100 μL of medium in 96-well plates (Corning Incorporated, Corning, NY, USA) and incubated for 24 h for attachment. Medium was removed and replaced with 100 μL serum-free DMEM medium containing different concentrations of Aβ42 oligomers ranging from 20 μM to 0 μM in order to determine the optimal Aβ42 oligomers concentration. To assess the effect of M5, M10, M29, and the wild type on the recovery of Aβ42-induced cell toxicity, different concentrations of these antibodies (0, 0.125, 0.25, 0.5 μM) were added to samples of 5 μM Aβ42 peptide. After 48 h incubation, CCK-8 assay was performed. The cell medium was replaced with DMEM, and CCK-8 reagent (Dojindo Laboratorise, Shanghai, China) (CCK-8:culture medium = 1:10, *v*/*v*) was added to each well, before the mixture was incubated at 37 °C in the dark for 0.5 h. The absorbance value was measured at 450 nm (reference wavelength at 600 nm) using a microplate reader (infinite M200 PRO; Tecan, Männedorf, Switzerland). Each condition consisted of three replicas per experimental group.

### 4.10. AL Fibrils Formation Measured by Thioflavin T Fluorescence

Synthetic light chain amyloid lyophilized peptide LN (1–30) was dissolved in PBS and 0.05% NaN_3_ and filtered with 0.2 μm membranes prior to use to remove formed aggregates. Peptides were prepared at a concentration of 0.2 mg/mL and added with different concentration of mAbs (2, 1, 0.5 μM). After continuous shaking (225 rpm) for 72 h at 37 °C, aliquots of samples were mixed with 10 μM thioflavin T (ThT, 97%; Macklin, Shanghai, China), followed by incubation at room temperature in the dark for 30 min. All fibril samples were loaded in a black-wall, clear-bottom 96-well half-area polystyrene microplate (Corning Incorporated, Corning, NY, USA) with a total volume of 50 μL per well, and measurements were performed in triplicate. ThT fluorescence was measured using plate reader (infinite M200 PRO; Tecan, Männedorf, Switzerland) with excitation at 450 nm and emission at 482 nm from the bottom of the well. For each experiment, control reactions using PBS were carried out, and the ThT fluorescence of antibodies incubated without AL fibrils was used as control.

### 4.11. AL Fibrils Bioactivity Measured by Toxicity on AC16 Cells 

Synthetic light chain amyloid lyophilized peptide LN (1–30) was dissolved in PBS, filtered with 0.2 μm membranes, and prepared at a final concentration of 50 μM in a sterile environment. Then, 100 μL of peptides was mixed with an equal amount of different concentrations of M5 (4, 0.4, 0.04 μM). After continuous shaking for 96 h at 37 °C, 225 rpm, samples were diluted 10 times with DMEM (Thermo fisher, Life Technologies, Carlsbad, CA, USA), adding 10% fetal bovine serum (Gibco, Carlsbad, CA, USA) for MTT assay.

The AC16 human cardiomyocyte cell line (American Type Culture Collection, ATCC) was cultured in DMEM medium with 10% fetal bovine serum (Gibco, Carlsbad, CA, USA) in 5% CO_2_ at 37 °C. The cells were seeded at a density of 5000 cells/well in 100 μL of medium in 96-well plates and incubated overnight. Medium was removed and replaced with 100 μL prepared samples. MTT assay was performed after a 72 h incubation. The culture medium was replaced with 100 μL serum-free DMEM, adding 10 μL of 5 mg/mL MTT stock solution (Yeasen, Shanghai, China). After 4 h incubation at 37 °C in the dark, the mixture was replaced by 100 μL stop solution, and the plate was incubated for another 4 h to solubilize the precipitates. The absorbance value was measured at 570 nm using a microplate reader. There were three replicas per experimental group.

### 4.12. Statistical Analysis 

Data analysis was performed using GraphPad Prism 5 software, and two-tailed unpaired Student’s *t*-test was used to compare two groups. For all experiments, *p*-values < 0.05 were considered statistically significant. Scatter dot plots depict the mean, with error bars representing the standard error of the mean (SEM).

## 5. Conclusions

In conclusion, we described a hypothetical concept that an amyloid antibody can be engineered by a few mutations to adapt to other amyloid sequences. The combination of structure-based modeling and experimental confirmation may provide a way to reposition and re-engineer therapeutic antibodies to target different amyloid diseases. The way forward is by aiming to increase specificity to the specific target, which could diminish toxicity. For this, additional mutation cycles appear to be the best vehicle.

## Data Availability

The data that support the findings of this study are available from the corresponding author upon reasonable request.

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
