# Peer review of "Re-Engineering Therapeutic Anti-Aβ Monoclonal Antibody to Target Amyloid Light Chain"

_ijms, 2024, doi:10.3390/ijms25031593_

Round 1

Reviewer 1 Report

Comments and Suggestions for Authors

I was initially very excited by the aims of this paper and initial results. However I didn't discover until the end of the paper that the study was not actually using a LC but instead a very small (and non-physiologically relevant) fragment. This therefore completely changes all of the claims made by the authors. If this study is to be published the use of a very small fragment of LC needs to be clear in the abstract and throughout the paper so that the reader can interpret the data knowing this. The authors themselves even state that the amyloid structures of κ light chain variable domain and 30 residue long LN peptide should be morphologically different!

There is also confusion between claiming that AL and AD could co-exist in the Introduction, but in the discussion the authors state that there is no correlation of these 2 diseases.

Figure 2 is very confusing to follow and Figure 4 is mis-labelled.

I did not have access to the Supplementary Information so cannot comment on any of that.

Initially I thought the mass spec analysis was going to tell me binding sites but it is actually only confirmation of correct expression of Abs therefore this could all be moved to Supplementary and discussed alongside SDS-PAGE data.

Why were 2 different toxicity assays used?

Additional suggestions and indications of some of the issues are highlighted by comments on the uploaded pdf.

Comments on the Quality of English Language

The language contains a mixture of present and past tense throughout and also many areas of poor grammar/missing words. A very detailed revision of language is needed before any consideration of publication. This includes use of abbreviations once defined.

Reviewer 2 Report

Comments and Suggestions for Authors

The manuscript entitled “Re-engineering therapeutic anti-Aβ monoclonal antibody to target amyloid light chain” by Bai et al has modified existing amyloid antibody Crenezumab to prepare its variants for generating effective therapeutic agents. The results are interesting and can be published with few minor changes.

Introduction section needs to be concise.

Axis labels in figure 5 is not legible. Authors should change those.

Reviewer 3 Report

Comments and Suggestions for Authors

The manuscript titled "Re-engineering therapeutic anti-Aβ monoclonal antibody to target amyloid light chain" focuses on the re-engineering of Crenezumab to target both Aβ oligomers and amyloid light chains (AL). It involves computational design and experimental validation, highlighting the creation and testing of multiple antibody variants, especially M5, M10, and M29. These variants demonstrated promising binding affinities and were analyzed through various techniques including molecular modeling, simulation, antibody engineering, SDS-PAGE, ELISA, and surface plasmon resonance (SPR) assays. The study emphasizes the potential of repurposing existing therapeutic antibodies for new targets, specifically in treating amyloidosis-related diseases like Alzheimer’s and systemic light chain amyloidosis. However, I have some concerns listed below that need to be addressed in the revised manuscript before it can be considered for possible publication in the IJMS.

1.      Figures in Manuscript and Original Gels/Blots: The manuscript includes detailed figures (e.g., Fig 3, Fig 4) showing results from SDS-PAGE, ELISA, and SPR assays. However, it doesn't explicitly address the use of the same figures in both sections. Raw data in supplementary materials should ideally correspond to these figures for transparency.

2.      Selection Criteria for Molecules: The selection process for antibody mutants is detailed, focusing on mutations enhancing binding to both Aβ oligomers and AL fibrils, based on computational and experimental validation.

3.      Clarity of Figures: Enhance chromatograms and other figures for clarity. More descriptive labels and legends are needed.

4.      Captions in Supplementary Material: Provide more comprehensive captions in the supplementary material to explain the significance of the findings.

Round 2

Reviewer 1 Report

Comments and Suggestions for Authors

The authors have mis-understood many of my comments and concerns with this manuscript and have not addressed the main issue in the revised manuscript in that the form of protein associated with AL used is only a small fragment of a light chain and this is not made clear to the reader until the very end of the paper. 

My second comment regarding the confusion between claiming that AL and AD could co-exist in the Introduction, but in the discussion the authors state that there is no correlation of these 2 diseases has also not been addressed in the response.

Neither has my comment about why 2 different toxicity assays were used (not why different cell types).

Comments on the Quality of English Language

English language has improved from first submission.

Round 3

Reviewer 1 Report

Comments and Suggestions for Authors

Despite me rejecting this manuscript it seems the authors were given the opportunity to respond so I am fine with publication.

Author Response

Thank you for your agree for publication. Thanks a lot.